# Revisiting Traditional Medicinal Plants: Integrating Multiomics, In Vitro Culture, and Elicitation to Unlock Bioactive Potential

**DOI:** 10.3390/plants14132029

**Published:** 2025-07-02

**Authors:** Erna Karalija, Armin Macanović, Saida Ibragić

**Affiliations:** 1Laboratory for Plant Physiology and Molecular Biology, Department for Biology, Faculty of Science, University of Sarajevo, Zmaja od Bosne 33-35, 71000 Sarajevo, Bosnia and Herzegovina; 2Centre for Ecology and Natural Resources “Academic Sulejman Redžić”, Department for Biology, Faculty of Science, University of Sarajevo, Zmaja od Bosne 33-35, 71000 Sarajevo, Bosnia and Herzegovina; armin.macanovic@pmf.unsa.ba; 3Department of Chemistry, Faculty of Science, University of Sarajevo, Zmaja od Bosne 33-35, 71000 Sarajevo, Bosnia and Herzegovina; saida.i@pmf.unsa.ba

**Keywords:** elicitors, phytochemicals, antioxidant activity, proteomics, synthetic biology, traditional medicine

## Abstract

Traditional medicinal plants are valued for their therapeutic potential, yet the full spectrum of their bioactive compounds often remains underexplored. Recent advances in multiomics technologies, including metabolomics, proteomics, and transcriptomics, combined with in vitro culture systems and elicitor-based strategies, have revolutionized our ability to characterize and enhance the production of valuable secondary metabolites. This review synthesizes current findings on the integration of these approaches to help us understand phytochemical pathways optimising bioactive compound yields. We explore how metabolomic profiling links chemical diversity with antioxidant and antimicrobial activities, how proteomic insights reveal regulatory mechanisms activated during elicitation, and how in vitro systems enable controlled manipulation of metabolic outputs. Both biotic and abiotic elicitors, such as methyl jasmonate and salicylic acid, are discussed as key triggers of phytochemical defense pathways. Further, we examine the potential of multiomics-informed metabolic engineering and synthetic biology to scale production and discover novel compounds. By aligning traditional ethnobotanical knowledge with modern biotechnology, this integrative framework offers a powerful avenue to unlock the pharmacological potential of medicinal plants for sustainable and innovative therapeutic development.

## 1. Introduction

Traditional medicine has long served as a cornerstone of healthcare systems worldwide, particularly in regions with limited access to synthetic pharmaceuticals. It includes a wealth of therapeutic knowledge with centuries of empirical observation, cultural heritage, and intergenerational transmission [1,2]. According to the World Health Organization, approximately 80% of the global population still relies on traditional medicine for primary healthcare needs [3,4], highlighting its continuing relevance.

Europe alone utilizes around 2000 medicinal and aromatic plant species commercially, with Germany accounting for approximately 1500 and Spain between 450 and 800 [5,6]. Globally, the WHO has catalogued more than 21,000 medicinal plant species, and more than half of modern pharmaceuticals are derived directly or indirectly from plant-based compounds [7]. Despite this, the therapeutic potential of many species remains underexplored, with a significant proportion of pharmacologically active constituents yet to be identified or mechanistically characterized [8].

A large portion of medicinal plants are harvested from wild ecosystems. These wild edible plants, which thrive without human cultivation, serve both nutritional and therapeutic roles in traditional societies [9]. The accumulated ethnobotanical knowledge held by local communities reflects an understanding of plant efficacy and safety, developed through generations of use [10,11,12,13]. Many such plants have been preserved and transmitted through oral traditions and daily practices, forming the basis for culturally embedded healing systems. Modern scientific exploration is increasingly validating this traditional knowledge base. Studies from southeastern Europe, for instance, have demonstrated that numerous traditionally used species, such as yarrow (*Achillea millefolium*), St. John’s wort (*Hypericum perforatum*), and nettle (*Urtica dioica*), exhibit high levels of secondary metabolites and significant antioxidant and antimicrobial activity [14,15,16,17,18]. These findings reveal statistically significant correlations between bioactive phytochemicals and their pharmacological effects, reinforcing the value of ethnopharmacological evidence.

The integration of multiomics technologies, particularly metabolomics, transcriptomics, and proteomics, has revolutionized the study of medicinal plants. These approaches allow for the comprehensive profiling of secondary metabolites, the research of biosynthetic pathways, and the discovery of novel biomarkers relevant to therapeutic use [19,20,21]. Recent works have used these techniques to explore stress-induced biochemical changes in plants such as lavender (*Lavandula angustifolia*), common grape wine (*Vitis vinifera*), and *Hypericum perforatum*, revealing stress-responsive shifts in flavonoids and other secondary metabolites [22].

In parallel, advances in in vitro culture systems, elicitor application, and co-cultivation models have created new possibilities for enhancing the yield of target compounds. Elicitation using jasmonic acid, beneficial microbes, or abiotic stressors has been shown to significantly increase the production of valuable metabolites such as ginsenosides, tanshinones, and flavonoids [23,24,25]. Moreover, nanotechnology and bioengineering approaches such as the use of biogenic gold nanoparticles synthesized with plant extracts (e.g., *Garcinia kola*) have shown promise in enhancing cytotoxic activity against cancer cells, potentially expanding the applications of traditional medicines into oncological domains [26].

Therefore, integrating traditional medicinal knowledge with modern multiomics platforms and biotechnological tools offers a strategic bridge between historical wisdom and contemporary pharmaceutical development. This multidisciplinary approach not only validates traditional practices but also provides scalable and sustainable methods for producing high-value bioactive compounds. The convergence of these methodologies is paving the way for a new era in drug discovery, one that is rooted in ecological knowledge, enhanced by technological precision, and driven by the need for accessible, natural therapeutics.

This review contributes by systematically integrating metabolomic, proteomic, and elicitation-based approaches to uncover and enhance phytochemical potential in underexplored medicinal plants. It introduces underutilized in vitro and biotic elicitation strategies, links multiomics data with bioactivity assays, and outlines future applications in synthetic biology and metabolic engineering. By doing so, it offers a practical roadmap for researchers aiming to design optimized culture systems, scale production of target compounds, and apply omics-guided interventions for therapeutic innovation.

This review highlights how such integrative strategies can systematically unlock the hidden pharmacological potential of traditional medicinal plants, transforming empirical knowledge into scientifically grounded, globally relevant therapeutic solutions.

## 2. Metabolomics: Chemical Profiling of Bioactive Compounds

In plant science, metabolomics has become essential for detailed analysis of plant components, and evaluation of quality, nutritional, and sensory properties, as well as investigation of functional and toxicological aspects [27]. The metabolome varies by species, with plants having a particularly rich diversity—more than 200,000 metabolites across the plant kingdom and 5000 to 25,000 in a single plant. For example, thale cress (*Arabidopsis thaliana*) produces around 5000 metabolites, significantly more than microorganisms (~1500) or animals (~2500) [28]. Due to the metabolome’s complexity, researchers typically use strategies that integrate multiple separation techniques with mass spectrometry, such as gas chromatography–mass spectrometry (GC–MS), liquid chromatography–mass spectrometry (LC–MS), nuclear magnetic resonance (NMR) spectroscopy, matrix-assisted laser desorption/ionization (MALDI), capillary-based MS, and other MS-based methods [29,30]. A preliminary method for separating and identifying phytochemicals, high-performance liquid chromatography (HPLC) is a leading technique for detecting plant-based metabolites [31]. Many pharmacopoeias of herbal medicine rely on HPLC fingerprinting to ensure consistency of plant extracts. Volatile and thermally stable compounds are typically analyzed using GC–MS, which provides both quantitative and qualitative data on essential oils, terpenoids, and other small volatiles found in herbs. GC-MS primarily uses two ionization methods: electron ionization (EI) and chemical ionization (CI), with EI being the most common in metabolomics. GC coupled with time-of-flight mass spectrometry (TOF-MS) is preferred for metabolic profiling due to its higher mass accuracy, faster acquisition times, and better deconvolution of complex mixtures [32]. On the other hand, non-volatile, thermolabile, and high-molecular-weight compounds that cannot be handled by GC are frequently analyzed using LC-MS/MS. This method is highly sensitive, and its throughput capacity makes it a “workhorse” for untargeted metabolomics in medicinal plant research [33]. LC-MS also uses different ionization methods and columns, with C18 or C8 reverse-phase columns being common. Mobile phases typically mix water with acetonitrile or methanol. Atmospheric pressure ionization (API) and electrospray ionization (ESI) are the main ionization techniques used [34]. While less sensitive, NMR provides structural information and is unmatched in interpreting novel compound structures. It is sometimes integrated into metabolomic workflows for dereplicating known compounds and confirming new ones [35,36].

Using these tools, researchers have catalogued the chemical profiles of numerous traditional plants. Untargeted metabolomics can generate a “metabolic fingerprint” of an herbal extract, which is essentially a chemical map of all detectable metabolites. By comparing metabolic fingerprints of a plant under different conditions (different growth stages, wild vs. cultivated, untreated vs. elicited, etc.), one can identify which compounds correlate with potent bioactivities or stress responses. Comprehensive metabolite profiling helps in dereplication, the process of recognizing known compounds early so that effort can focus on novel constituents. This prevents “rediscovery” of already known chemicals and fast-tracks the identification of unique bioactive molecules.

Major bioactive classes characterized through metabolomics include alkaloids, terpenoids, phenolics (such as flavonoids and tannins), and glycosides, each associated with distinct biological activities [24,27,30]. Alkaloids often contribute to neuroactive and antimicrobial effects; terpenoids exhibit broad-spectrum antimicrobial, anti-inflammatory, and anticancer properties, while phenolic compounds are key contributors to antioxidant activity and immune modulation [26].

## 3. Proteomics: Insights into Pathways and Mechanisms

Proteomics has become a cornerstone methodology in research focused for investigation of biological mechanisms underlying the bioactivity of plant-derived metabolites. By investigating the complete protein complement of a plant cell or tissue under defined conditions, proteomic analyses allow researchers to identify enzymes directly involved in the biosynthesis of secondary metabolites, as well as regulatory proteins that modulate these biosynthetic pathways [24,30].

Medicinal plants are known for producing a wide array of bioactive compounds, many of which play a significant role in traditional medicine and drug development [1,3]. However, the precise molecular pathways governing the synthesis and action of these metabolites often remain poorly understood. Proteomic tools such as two-dimensional gel electrophoresis (2-DE), liquid chromatography coupled with tandem mass spectrometry (LC-MS/MS), and isobaric tags for relative and absolute quantitation (iTRAQ) are pivotal for identifying and quantifying proteins that correlate with the accumulation of specific phytochemicals [24,27].

Recent studies have demonstrated the importance of proteomics in mapping secondary metabolic pathways. For instance, in Asian ginseng (*Panax ginseng*), treatment with methyl jasmonate resulted in elevated levels of oxidosqualene cyclase (OSC) and S-adenosylmethionine synthase (SAMS), enzymes crucial in ginsenoside biosynthesis [24] (Table 1). Similarly, in English jew (*Taxus baccata*), 10-deacetylbaccatin III-10-O-acetyltransferase and T5αH were upregulated in response to MeJA, correlating with taxol biosynthesis [24]. Additionally, Liu et al. [20] used comparative proteomics to investigate different species and successfully identified species-specific enzymes and metabolic proteins associated with saponin biosynthesis. Such findings not only help in understanding interspecies variation but also in selecting elite cultivars for pharmaceutical applications.

In addition to pathway investigations, proteomics contributes to understanding the biological activities of metabolites by identifying their molecular targets and assessing their mode of action at the cellular level. Proteomic approaches have been employed to study the interactions of bioactive compounds with specific enzymes, receptors, and signaling proteins, revealing how these metabolites exert antioxidant, antimicrobial, or anti-inflammatory effects [26,31]. For example, targeted proteomic analysis of *Arabidopsis thaliana* treated with a phenolic-rich extract of *Hypericum perforatum* revealed upregulation of glutathione S-transferase (GSTs), heat shock proteins (HSP70), and pathogenesis-related proteins (PR-2, PR-5), indicating their involvement in oxidative stress mitigation and immune priming [25,32].

Numerous studies have used comparative and targeted proteomics to identify key biosynthetic enzymes and regulatory proteins in elicited medicinal plants. For example, β-amyrin synthase, CYP88D6, and UGT73C11 were upregulated in liquorice (*Glycyrrhiza glabra*) following MeJA treatment, enhancing glycyrrhizin biosynthesis [24]. In Asiatic pennywort (*Centella asiatica*), MeJA elicitation increased expression of enzymes in the triterpenoid saponin pathway, including OSC and several UDP-glycosyltransferases (UGTs). These findings emphasize how proteomic approaches can reveal not only the metabolic enzymes but also regulatory nodes that respond to elicitation stimuli.

Multiomics integration, where proteomics is combined with transcriptomics and metabolomics, enables the construction of comprehensive biosynthetic networks and regulatory frameworks [28,30]. This systems-level approach allows for the identification of rate-limiting steps, feedback loops, and transcriptional regulators that are essential for the coordinated production of secondary metabolites [24,30,32]. Furthermore, proteomics is important for validating candidate genes and enzymes discovered through genomics and transcriptomics. It provides direct evidence of protein expression and post-translational modifications, which are often critical for enzyme activity and metabolite specificity [30,32]. Advances in bioinformatics tools and protein databases have significantly enhanced the accuracy of protein identification and functional annotation, especially in non-model medicinal plants [27,28]. Commonly used databases such as UniProt and PRIDE enable researchers to identify conserved protein sequences and access publicly available proteomics data. Tools like Blast2GO and InterProScan support functional annotation by mapping protein sequences to gene ontology (GO) terms and protein families. Additionally, KEGG and STRING databases facilitate the visualization of metabolic and protein–protein interaction pathways, respectively, allowing for system-level insights. However, challenges remain in applying these tools to medicinal plants due to limited genomic resources, incomplete annotations, and cross-species transfer errors, which can affect pathway reconstruction and result interpretation. Despite these limitations, continuous improvements in computational tools and growing datasets are progressively closing this gap.

## 4. Integrating Phytochemical Profiles with Biological Activity Assays

A crucial aspect of natural products research is connecting chemical profiles with biological activities. In the area of medicinal plants, two widely studied activities are antioxidant capacity and antimicrobial effects. Researchers often use in vitro assays to evaluate these properties and then correlate the results with metabolomic data to identify active compounds or synergistic combinations.

### 4.1. Antioxidant Activity Assays

Medicinal plant extracts are rich in phenolics, flavonoids, and other antioxidants that can neutralize harmful free radicals. To quantify this capacity, colorimetric assays like DPPH, ABTS, and FRAP are commonly employed. The DPPH assay measures the ability of an extract or compound to scavenge the stable free radical DPPH (2,2-diphenyl-1-picrylhydrazyl), which has a deep purple color. When an antioxidant is present, it reduces DPPH, causing a change in color that can be quantified spectrophotometrically [43]. Similarly, the ABTS assay involves generation of a blue green ABTS•^+^ radical cation that is reduced to a colorless form by antioxidants; the extent of decolorization (usually measured at 734 nm) reflects the sample’s total antioxidant power [44]. The FRAP test is a standard single electron transfer-based method that measures the ability of antioxidants to reduce ferric ion (Fe^3+^)-ligand to the blue-colored ferrous complex (Fe^2+^) in acidic conditions [45]. These assays are straightforward, high-throughput, and commonly used to evaluate the antioxidant activity of medicinal plants [44]. For example, extracts of acacia. (*Vachellia*) species have been compared via DPPH and ABTS assays to determine which has the highest radical scavenging activity, correlating those results with the abundance of phenolic compounds in each extract [46]. Such correlations often show that extracts with higher total phenolic or flavonoid content exhibit stronger DPPH/ABTS scavenging (lower IC_50_ values), consistent with the known antioxidant role of these phytochemicals [44].

Literature research revealed an innovative quantitative antioxidant activity assay based on cellular metabolomics, named the cellular metabolomics antioxidant activity (CMAA) assay. Fu et al. [47] analyzed changes in cellular metabolites in HepG2 cells exposed to oxidative stress. The method employs multivariate analysis techniques such as PCA and OPLS-DA, using vitamin E as a standard to quantify antioxidant effects. The CMAA assay was applied to evaluate the antioxidant activity of flavonoids extracted from Maltese mushroom (*Cynomorium songaricum*), demonstrating its potential as a powerful tool for metabolomics-based antioxidant assessment.

Looking ahead, antioxidant activity assays in metabolomics are likely to become more precise, faster, and better suited to real biological systems. With improvements in technology, future methods may allow real-time tracking of metabolic changes in cells, giving a clearer picture of how antioxidants work. This could lead to better ways of identifying effective natural compounds and understanding their roles in health, especially in areas like drug development and nutrition science.

### 4.2. Antimicrobial Activity Assays

To investigate the ability of plant extracts or isolated compounds to inhibit microbial growth, researchers frequently use agar diffusion methods and broth dilution methods. The classical agar disc diffusion (or agar well diffusion) assay is a qualitative method where a filter paper disc soaked with the plant extract is placed on an agar plate inoculated with a test bacterium. If the extract has antimicrobial properties, it will create a clear zone of inhibition around the disc where bacterial growth is prevented. This method is one of the oldest and most common antimicrobial screening techniques, valued for its simplicity and low cost [48].

While disc diffusion provides a yes/no indication and relative potency (via zone size), the minimum inhibitory concentration (MIC) is determined by broth dilution assays. In a MIC assay, the extract is diluted in a liquid growth medium inoculated with the target microbe to find the lowest concentration that completely inhibits visible growth. Together, disc diffusion and MIC assays allow researchers to gauge the antimicrobial spectrum and strength of medicinal plant extracts. These assays have been applied to numerous traditional plants, for instance, testing neem (*Azadirachta indica*) leaf extracts against *Staphylococcus* and *Escherichia coli*, or evaluating combinations of herbs in traditional formulations against multi-drug-resistant pathogens. By linking these bioassay results with metabolomic findings, scientists can pinpoint which components of an extract are responsible for antimicrobial effects. Often, metabolite–bioactivity correlation will spotlight a few candidate compounds (like alkaloids or terpenes) that strongly correlate with larger inhibition zones or lower MIC values. Those compounds become prime targets for isolation and further testing, potentially yielding new antibiotic leads. In addition to that, characterizing microbial metabolism could pave the way for discovering new antimicrobial drug targets and treatments. The metabolic state of bacterial cells during antibiotic treatment can influence or result from antimicrobial resistance (AMR). Reduced metabolic activity can decrease antibiotic uptake, while increased activity supports energy-demanding AMR mechanisms like cell-wall modifications and efflux pump overexpression. Understanding these metabolic processes may help strategically alter metabolism during treatment to resensitize pathogens to antibiotics [49].

As an example, Hoerr et al. [35] hypothesized that combining intracellular metabolic fingerprints and extracellular footprints can offer a more complete understanding of novel antibiotic mechanisms in drug discovery. Using 1H NMR spectroscopy, they analyzed the metabolic responses of *Escherichia coli* to various antibiotics, finding distinct fingerprints and footprints that revealed information on intracellular and extracellular targets, and successfully predicted the modes of action of several antibiotics. Further on, metabolomics plays a crucial role in developing alternative antimicrobial therapies by exploring mechanisms like quorum sensing (QS), which regulates microbial communication and processes such as virulence factor production and biofilm formation. With increasing antibiotic resistance, quorum quenching (QQ) is being studied to disrupt microbial communication and inhibit these harmful processes. Techniques such as chromatography–mass spectrometry, bioluminescence, fluorescence, electrochemistry, and colorimetry enable both qualitative and quantitative measurement of QS/QQ molecules, providing valuable insights for alternative antimicrobial strategies [50].

### 4.3. Integrative Analysis: Linking Metabolites, Proteins, and Bioactivity

This integrative strategy, often referred to as biochemometric analysis, combines chemical fingerprinting with biological response data to identify metabolite patterns that correlate with specific therapeutic activities, such as antioxidant, antimicrobial, or anti-inflammatory effects.

Multivariate statistical tools such as principal component analysis (PCA), partial least squares discriminant analysis (PLS-DA), and orthogonal PLS-DA (OPLS-DA) are often applied to analyze these complex datasets. Through this analysis it is possible to identify clusters of metabolites whose abundance consistently tracks with bioactivity across multiple samples or conditions. For example, metabolites that show high correlation with antioxidant activity can be visualized in score plots or heatmaps, highlighting compound classes responsible for radical scavenging effects [29,51]. Candidate metabolites identified through biochemometric models can then be prioritized for purification, structural elucidation, and functional validation. Studies using this approach have successfully narrowed down complex extracts to a handful of key contributors. In *Hypericum perforatum*, for instance, integration of metabolomic and bioactivity data revealed that compounds such as hyperforin and quercetin glycosides were consistently linked to antioxidant and antimicrobial potency [22].

When plants demonstrate strong bioactivity, such as antimicrobial or anticancer effects, parallel upregulation of defense-related proteins, including pathogenesis-related (PR) proteins, heat-shock proteins, or enzymes involved in oxidative stress responses, may be observed. Such findings support the hypothesis that metabolite–protein interactions or co-regulation networks underlie the observed biological outcomes. These integrative studies provide insights not only into bioactivity but also into the ecological and physiological context of metabolite production [43,52].

Cluster analyses and correlation matrices can also be applied to group metabolites with similar activity profiles. When paired with hierarchical heatmaps comparing metabolite intensities and measured bioactivities (e.g., DPPH scavenging, MIC values), these tools highlight both known and novel contributors to pharmacological effects. The result is a refined shortlist of bioactive compounds or compound classes that warrant further investigation.

This integrated, data-driven pipeline significantly advances natural product research. It enables rational prioritization over random screening and aligns with modern drug discovery frameworks. By leveraging multiomics in this way, researchers can systematically connect chemical diversity with therapeutic function, thereby enhancing the efficiency and relevance of studying traditional medicinal plants. Key classes of active compounds, including alkaloids, terpenoids, phenolics, and flavonoids, exhibit well-documented pharmacological effects such as antioxidant, antimicrobial, anti-inflammatory, and anticancer activities [24,26,43,46,49]. For example, phenolic compounds like chlorogenic acid and rutin contribute strongly to antioxidant activity, while alkaloids (e.g., scopolamine in Egyptian henbane, *Hyoscyamus muticus*) and terpenoids (e.g., ginsenosides in *Panax ginseng*) display notable antimicrobial and cytotoxic properties [26,46,49]. By correlating these compound classes with observed bioactivities, integrative multiomics approaches further elucidate the therapeutic potential of traditional medicinal plants.

## 5. In Vitro Culture Systems as Platforms for Bioactive Compound Production

Harnessing the full potential of medicinal plants often requires obtaining enough of their bioactive compounds, which can be challenging due to slow growth, low natural abundance of the compounds, or environmental constraints. In vitro plant culture systems offer an attractive solution by providing controlled, sustainable production platforms for secondary metabolites. The concept of using “plant cell factories” involves growing plant cells, tissues, or organs (like roots or shoots) under sterile laboratory conditions, where variables can be controlled to optimize yield [24].

Different types of in vitro culture can be used for the purpose of obtaining bioactive compounds (Table 1):Callus cultures: Masses of undifferentiated plant cells induced from explants on agar media. Callus can be proliferated indefinitely and later induced to form other structures or specialized cells;Cell suspension cultures: Cells from callus that are dispersed and grown in liquid shake flasks or bioreactors. These are particularly useful for metabolite production as they can be scaled up and are amenable to elicitor addition in a uniform way;Hairy root cultures: Fast-growing, genetically stable root cultures obtained by infecting a plant with *Agrobacterium rhizogenes*. Hairy roots often exhibit high productivity of root-derived secondary metabolites (e.g., alkaloids in *Catharanthus* or ginsenosides in *Panax*);Organ cultures (shoots, somatic embryos, etc.): Sometimes specific plant organs are cultured to exploit organ-specific pathways (for instance, shoot cultures of sweet sagewort (*Artemisia annua*) to produce artemisinin, since leaves produce this compound);

In vitro cultures present several advantages over field-grown plants. They are not subject to seasonal or climate variability, and they eliminate the need to harvest wild plants (many of which may be endangered or slow-growing) [24]. For rare medicinal plants that are hard to cultivate or take years to mature, in vitro culture might be the only viable option to obtain their valuable metabolites in an eco-friendly manner [53]. A classic example is *Taxus* sp., the source of the anti-cancer drug paclitaxel. Wild trees grow slowly, and paclitaxel content is low, but cell suspension cultures of *Taxus* have been developed that produce this compound on a commercial scale, augmented by elicitation strategies. Several pharmaceutical companies produce paclitaxel using plant cell culture technology, demonstrating that “cell factory” approach at industrial scale [54].

However, simply establishing a plant cell culture is often not enough. Yields of secondary metabolites can initially be low because cultured cells may produce less of the specialized compounds than their wild counterparts. This is where elicitation (discussed in the next section) becomes critical. By adding certain stimuli or elicitors to the culture, one can mimic stress conditions that trigger the cells to ramp up their defense-related secondary metabolism [55].

In vitro systems are also ideal for experimental manipulation and multiomics analysis. Because the environment is controlled, researchers can systematically apply different treatments (like varying nutrients, hormones, or stress factors) and then use metabolomics and proteomics to observe the changes (Table 2). For example, one can grow parallel flasks of a plant cell suspension, treat some with a signaling molecule (elicitor) and leave others untreated, and then compare their metabolite profiles and protein expression. The differences will highlight which pathways are activated by the treatment. This level of control and replication is hard to achieve in whole plants in the field or greenhouse. Thus, in vitro cultures serve as modular platforms for both production and scientific discovery; they allow enhancement of bioactive compound yields and enable deep multiomics insights into how those enhancements occur [56].

Secondary metabolites induced by elicitors, including alkaloids, terpenoids, and phenolics, play critical roles in the plant’s defense mechanisms and contribute to therapeutic properties such as antimicrobial, antioxidant, and anti-inflammatory effects [24,26,27,43,49]. The selective induction of these compound classes through targeted elicitation can thus enhance the pharmacological value of in vitro cultures.

### 5.1. Elicitation Strategies: Awakening the Phytochemical Defense

Plants do not produce high levels of secondary metabolites without reason: these compounds often serve as defenses against pests, pathogens, or abiotic stress. Elicitors are signals that provoke a plant’s phytochemical defenses, stimulating the production of secondary metabolites. In biotechnology, elicitation has become a great strategy to boost yields of valuable compounds in plant cell and tissue cultures [24]. An elicitor can be defined as a compound or factor, introduced in small amounts, that triggers a defense or stress response leading to enhanced biosynthesis of target metabolites [62]. Elicitors are broadly categorized into biotic (of biological origin) and abiotic (physical or chemical factors) [24,63]:Biotic elicitors include molecules associated with pathogens or beneficial microbes (e.g., fungal cell wall fragments like chitosan or glucans, bacterial lysates, yeast extract, and even whole microbial cells). These mimic an attack, causing the plant cells to display a defense response [64]. Some biotic elicitors can also be endogenous, meaning produced by the plant itself in response to stress, such as plant hormone signals;Abiotic elicitors include physical factors like light (UV radiation), temperature extremes, osmotic stress (high salt or drought simulation), and chemical factors like heavy metals or plant hormones that are not directly of microbial origin. These factors also induce stress responses that converge on secondary metabolism upregulation [65];

#### 5.1.1. Biotic Elicitors: Harnessing Biotic Signals to Boost Secondary Metabolite Production

Biotic elicitors, derived from microorganisms or plant-associated biomolecules, represent a powerful and sustainable tool to enhance the biosynthesis of secondary metabolites in medicinal plants. These elicitors include bacterial lysates, fungal extracts, yeast components, cell wall fragments (e.g., chitin, glucans), or even entire microbial cells. When introduced to plant cells or in vitro cultures, they mimic pathogen-associated molecular patterns (PAMPs), triggering plant immune responses that activate defense-related metabolic pathways. This response often results in elevated levels of bioactive compounds such as flavonoids, alkaloids, terpenes, and phenolics, compounds with demonstrated antioxidant, antimicrobial, and anticancer effects [66].

In other words, numerous plant-derived natural products are regularly used as therapeutic agents due to their diverse pharmacological properties. For instance, the cytotoxicity results emphasize the significant anticancer potential of specific phenolic compounds such as gallic acid, chlorogenic acid, p-coumaric acid, quercetin, and rutin, extracted from the leaves of yellow oleander (*Thevetia peruviana*), showing activity against human colon carcinoma and other cancer types [67]. Among flavonoids, compounds such as quercetin, hesperetin, and naringin have shown antiviral properties. Several studies have highlighted the antiviral activity of apigenin, vitexin, and their derivatives against a range of viruses, including hepatitis A, B, and C viruses, herpes simplex virus 1 (HSV-1), rhesus rotavirus (RRV), and influenza viruses. Certain flavonoids, including apigenin, galangin, and chalcone, also possess antibacterial properties. Moreover, delphinidin, an anthocyanin, has been shown to promote endothelium-dependent vasorelaxation, supporting its role in cardioprotection [68]. Terpenes, widely distributed in plants such as citrus fruits, pine trees, and mint, contribute not only to characteristic aromas but also to biological functions such as light harvesting, membrane stabilization, and hormone regulation. Their therapeutic potential includes antiplasmodial, antiviral, antidiabetic, neuroprotective, and analgesic activities [69]. A prominent example is artemisinin, a sesquiterpene lactone derived from *Artemisia annua*, whose bioactivity has been proven in the treatment of malignant cerebral malaria caused by *Plasmodium falciparum*. Another important terpene, isolated from Pacific jew (*Taxus brevifolia*), is used as an antimitotic agent in the treatment of ovarian and breast cancers, and has since been approved for various other cancer therapies [70]. Equally significant are alkaloids, which include pharmacologically active compounds such as reserpine, ajmaline, and serpentine, traditionally used to manage epilepsy, hypertension, and cardiac disorders. Although many alkaloids can be cytotoxic at high concentrations, substances like morphine, quinine, and caffeine remain essential drugs when administered at therapeutic doses [71]. One notable example is galantamine, an alkaloid derived from green snowdrop (*Galanthus woronowii*), which was approved by the FDA for the treatment of Alzheimer’s disease [70].

In terms of elicitation, in *Agrobacterium rhizogenes* elicitation of *Hypericum perforatum* cell suspensions [72] led to a significant increase in xanthone production. The bacterium acted as a biotic stress signal, activating pathways associated with phenolic metabolism, thereby enhancing the accumulation of compounds known for their antioxidant and cytotoxic properties. This underscores the potential of microbial elicitation not only as a metabolic trigger but also as a biotechnological strategy for optimizing yields of pharmacologically important molecules under controlled conditions.

Yeast extract (0.1% *w*/*v*) has also been used for enhancement of secondary metabolite production, where it was applied to Sarajevos’ widow flower (*Knautia sarajevensis*) in vitro shoot cultures to evaluate its effect on secondary metabolite accumulation and antioxidant activity [42]. The treatment resulted in a significant increase in total phenolics and flavonoids, with notably higher levels of chlorogenic acid and rutin compared to the control. Antioxidant capacity, assessed via the DPPH assay, was also significantly improved. These findings demonstrate that yeast extract acts as an effective biotic elicitor, enhancing the biosynthesis of phenolic compounds and improving the antioxidant potential of *K. sarajevensis* shoot cultures.

Biotic elicitors offer a unique advantage over chemical stimuli due to their complex and multifaceted nature, which allows them to simultaneously trigger multiple defense pathways. This complexity can lead to a more pronounced and sustained upregulation of secondary metabolism compared to abiotic elicitors. Furthermore, co-culturing approaches, where plant cells are grown in the presence of non-pathogenic bacteria or fungi, have shown promise for boosting metabolite yields through sustained signaling interactions. For instance, co-culture of red sage (*Salvia miltiorrhiza*) roots with *Bacillus* species significantly enhanced tanshinone accumulation, as revealed through targeted metabolomic analyses [73].

From an integrative multiomics perspective, the application of biotic elicitors is particularly powerful. By combining transcriptomics, proteomics, and metabolomics, researchers can map the full cascade of responses triggered by biotic stimuli, from receptor activation to secondary metabolite accumulation. This systems-level view provides insights into regulatory networks, transcription factors (e.g., WRKY, MYB, and AP2/ERF), and metabolic bottlenecks that can be manipulated to further enhance bioactive compound production [74].

#### 5.1.2. Abiotic Elicitors: Methyl Jasmonate and Salicylic Acid

Different abiotic factors can be used as eliciting agents but plant hormones have also been shown to have eliciting abilities. Methyl jasmonate (MeJA) and salicylic acid (SA) are among the most used as plant signaling molecules that regulate distinct branches of the plant defense network. Additionally, other pant hormones can also be used as elicitors such as citokinins. Research on the influence of cytokinin, 6-benzyladenine (BA), zeatin (ZEA), and kinetin (KIN), on the growth, phenolic compound accumulation, and biological activities of *Knautia sarajevensis* shoot cultures (Figure 1) [75] demonstrated significant eliciting effect. Among the compounds investigated, zeatin notably enhanced both biomass production and the accumulation of phenolic compounds, including salicylic acid, rosmarinic acid, and 4-hydroxybenzoic acid. These phenolics are recognized for their antioxidant and antimicrobial properties. Additionally, shoots cultivated with BA exhibited moderate antimicrobial activity against *Staphylococcus aureus* and *Bacillus spizizeni*. The findings suggest that cytokinins can serve as effective elicitors in plant tissue cultures, promoting the production of valuable secondary metabolites with potential therapeutic applications.

MeJA and SA act as endogenous “alarm” hormones, they activate molecular pathways in response to biotic and abiotic stresses. In plant tissue cultures, exogenous application of these hormones is a widely used strategy to stimulate the biosynthesis of pharmaceutically relevant secondary metabolites [76].

MeJA is among the most widely used elicitors due to its potent and broad-spectrum action. It initiates a rapid and multifaceted signaling cascade through receptor complexes such as COI1-JAZ, triggering the transcription of jasmonate-responsive genes. This cascade results in the biosynthesis of a wide array of secondary metabolites, particularly those involved in plant defense such as alkaloids, terpenoids, and phenolics.

Numerous studies have demonstrated MeJA’s efficacy in enhancing metabolite yields. For instance, *Panax ginseng* cultures exposed to MeJA exhibit a more than 10-fold increase in ginsenosides, while *Taxus* spp. show markedly elevated paclitaxel production [77]. MeJA also enhances the biosynthesis of podophyllotoxin in flax (*Linum*), scopolamine in *Hyoscyamus*, resveratrol in *Vitis vinifera*, and artemisinin in *Artemisia annua*. Mechanistically, MeJA induces early ROS production and ion fluxes, followed by transcription factor activation and downstream biosynthetic gene expression [78]. These molecular events culminate in metabolite accumulation within days. Integrated omics studies further validate MeJA’s impact. In Santa Catalina indian paintbrush (*Castilleja tenuiflora*), for example, MeJA not only increased phenolic compound content but also upregulated multiple genes within the phenylpropanoid pathway, establishing a clear link between elicitor treatment and biosynthetic output [79].

In contrast, SA plays a more targeted role, primarily modulating plant responses to pathogenic stress. It activates systemic acquired resistance (SAR) and is associated with the induction of pathogenesis-related (PR) proteins and phenolic compounds via the phenylpropanoid pathway. When applied to in vitro cultures, SA selectively boosts the production of specific bioactive metabolites [80].

In maidenhair tree (*Ginkgo biloba*), SA elicitation induced the accumulation of sesquiterpene trilactones (ginkgolides and bilobalide), while in *Knautia sarajevensis* shoot cultures, SA treatment led to significant increases in total phenolics and flavonoids, accompanied by enhanced antioxidant activity [42]. These responses are mediated through key enzymes such as phenylalanine ammonia-lyase (PAL), highlighting SA’s role in stimulating antimicrobial and antioxidant defense.

MeJA and SA can act also synergistically. Combined or sequential elicitation has been explored as a strategy to activate complementary pathways, resulting in more robust or diversified metabolic outputs. This elicitor crosstalk, where SA may prime plant tissues for enhanced MeJA responsiveness, or vice versa, has proven valuable in optimizing metabolite production systems. Both MeJA and SA are straightforward to apply in culture systems, typically at micromolar concentrations, and their effects are observable within 24–72 h. Beyond metabolite accumulation, transcriptomic and proteomic analyses reveal consistent upregulation of genes encoding key biosynthetic enzymes (e.g., terpene synthases, cytochrome P450s, glycosyltransferases), underscoring the depth and specificity of their action [24,81].

### 5.2. Mechanisms of Elicitor Action and Multiomics Linkages

At the cellular level, elicitor perception and response are a highly coordinated process (Figure 2). The multiomics approach allows us to capture this process at different molecular layers. For example, using transcriptomics or proteomics, one can observe the activation of specific MAPKs or the accumulation of specific TFs post-elicitation. Metabolomics then shows the outcome in terms of increased metabolites. Multiomics datasets can be integrated to form causal links: e.g., MeJA treatment → activation of bHLH transcription factor (proteomics identifies it) → upregulation of phenylpropanoid enzymes (proteomics/transcriptomics) → increased flavonoid content (metabolomics confirms) [82]. In the study on Asian water plantain (*Alisma orientale*), proteomic analysis of MeJA-elicited cultures revealed upregulation of several transcription factors (bHLH, MYB, etc.), correlating with increased biosynthesis of protostane triterpenes [83]. This kind of finding is valuable because it not only shows the effect (more product) but also the mechanism (which genes/proteins were responsible for the increase), pointing to targets for genetic engineering.

Elicitation strategies, whether using signaling molecules like MeJA/SA or applying abiotic stresses, are essential tools to unlock the full biosynthetic potential of medicinal plant cultures (Table 3). They push the plant cells to produce more of the compounds that might only be made under stress in nature. The integration of multiomics in studying elicitation has been revolutionary: it provides a comprehensive picture of how an elicitor’s signal travels from perception at the cell membrane (proteomic detection of receptor and kinase changes) to gene expression in the nucleus (transcriptomics/proteomics of TFs) to metabolite accumulation in the cell (metabolomics of end products). This holistic understanding guides smarter use of elicitors and even the design of new ones (such as synthetic analogues like coronatine, a potent JA-mimic, or novel light regimes) to maximize yields of bioactive compounds.

### 5.3. Multiomics Integration: Linking Elicitation to Metabolic Pathways

By combining metabolomics and proteomics (and often transcriptomics) in studies of medicinal plants, researchers can link cause and effect in the plant’s responses. Specifically, under elicitation or other treatments, multiomics reveals how shifts in gene/protein expression correspond to shifts in metabolite profiles. When applied in the context of in vitro culture systems, this multiomics integration allows for iterative optimization. By identifying which elicitor induces which molecular cascade, researchers can fine-tune culture conditions and timing of treatments for maximal yield. For instance, by pairing elicitation trials with real-time omics monitoring, it is possible to establish cause–effect relationships between elicitor stimuli and downstream metabolite accumulation. This offers a powerful strategy not just for profiling but for predictive improvement of metabolite production.

One concrete benefit of integration is the ability to construct biosynthetic pathway networks. For a given bioactive compound, metabolomics might show its increase upon elicitation, while proteomics could simultaneously show increases in several enzymes that belong to the pathway producing that compound. For example, in MeJA-elicited creaat (*Andrographis paniculata*)—a medicinal herb known as the diterpenoid andrographolide, an integrated analysis found that certain MEP-pathway enzymes (for terpenoid biosynthesis) were upregulated at the protein level, correlating with higher accumulation of andrographolide and related diterpenoids [82]. Such observation strongly suggests those enzymes are bottlenecks or rate-limiting steps now relieved by elicitation providing a valuable insight if one were to engineer that pathway.

Multiomics also helps in discovering new links and metabolites. This approach has been used to propose functions for previously unannotated enzymes. In basil (*Ocimum basilicum*) cell cultures treated with elicitors, researchers noted the emergence of new peaks in the LC-MS chromatogram corresponding to unknown phenolic conjugates; proteomic analysis revealed induction of a BAHD acyltransferase enzyme, suggesting it catalyzed the formation of those new phenolic conjugates [84]. Thus, multiomics can lead not only to finding new compounds, but also to assigning biosynthetic roles to genes/proteins—essentially piecing together new metabolic pathways. Another important aspect is understanding regulatory networks. By overlaying proteomic data (which might include signaling proteins, kinases, or transcription factors) with metabolite data, we can identify key regulators. For instance, the identification of ORCA2 in periwrinkle (*Catharanthus* sp.) was facilitated by examining the gene expression changes under elicitation [82]. New techniques like mass spectrometry imaging (spatial metabolomics) can show the location of metabolites in plant tissues [30], and when combined with spatial proteomics or transcriptomics (e.g., laser-capture microdissection followed by sequencing), it is possible to see where certain pathways are active in the plant.

From an analytical perspective, integrating datasets poses challenges, but also has dedicated solutions. Correlation analyses, network modeling, and machine learning are increasingly applied to multiomics data to extract meaningful patterns. For example, WGCNA (weighted gene co-expression network analysis) can integrate metabolite and protein expression profiles to find modules of co-regulated features—perhaps grouping a set of metabolites with a set of enzymes and a transcription factor that all vary together across different conditions [85]. Such a module could represent coordinated pathway activation. In one multiomics study of black cumin (*Nigella sativa*) under elicitation, a network analysis linked changes in thymoquinone levels (a bioactive compound) with changes in specific proteins of terpene metabolism and stress-response proteins, suggesting a network of genes that could be manipulated to enhance thymoquinone yield.

## 6. Gaps in Current Knowledge and Challenges

Despite significant advances, there remain significant gaps and challenges in fully realizing the potential of multiomics for traditional medicinal plants. Recognizing these gaps is important to chart future research directions.

Lack of genomic databases for non-model plants: Many medicinal plants lack fully sequenced or well-annotated genomes. A high-quality genome is the scaffold on which proteomic and transcriptomic data are interpreted. The absence of genomic information makes it difficult to identify proteins from mass spectra or to allocate functions to genes. This is a particular issue for proteomics: many proteins detected in a medicinal plant extract might be “unknown” because the sequence databases are incomplete. For instance, the diversity and complexity of plant proteomes and specialized metabolites can complicate protein identification, and many proteins have no known function or annotation [86]. This means multiomics studies might generate lists of changed proteins or transcripts, but not all can be linked to pathways due to our limited knowledge, leaving some metabolite changes unexplained.

Computational challenges in multiomics integration: Multiomics produces big data. Integrating datasets from metabolomics, proteomics, transcriptomics, etc., is computationally intensive and methodologically complex. Differences in data scaling, the sheer number of variables, and the dynamic range of each data type present obstacles. As a result, managing extensive multiomics data and extracting biologically meaningful information remains a challenge, requiring advanced bioinformatics and statistics [87]. There is a need for better algorithms and pipelines to correlate multi-layered data and to handle the “noise” inherent in each layer. Standardizing protocols across labs is also an issue—different metabolomic methods or growth conditions might yield data that are hard to compare. Without standardization, constructing comprehensive public databases or meta-analyses for medicinal plant omics is difficult [88].

Temporal and context-specific metabolic variation: Plant metabolism is not static; it fluctuates with development stages, circadian rhythms, and micro-environmental conditions. An elicitor might have a strong effect at one time-point and a diminished or different effect at another. Capturing these dynamics is challenging, as most studies take snapshots (e.g., 24 h after treatment). This can miss transient changes or feedback regulation. Additionally, metabolism can divert flux among multiple pathways; sometimes increasing one compound might unintentionally reduce others due to competition for precursors. Approaches to monitor real-time changes (perhaps through time-course multiomics) are needed to fully grasp the kinetic responses. The dynamic and context-dependent nature of specialized metabolism means results from a controlled lab setting might not fully predict what happens in a whole plant in nature [88,89].

Gaps between metabolite discovery and pharmacological validation: While multiomics gives molecular detail, linking that back to actual health benefits or pharmacological effects is still a gap. For example, an omics study might find new terpenoids in a plant, but are those terpenoids bioactive in a relevant assay or organism? There is often a disconnect between the discovery of metabolites and testing them for specific medicinal effects (anti-cancer, anti-diabetic, etc.). Bridging this gap will require more bioactivity-guided omics, where omics data is paired with functional assays at every step. This integration with pharmacology and systems biology of the human target is still in its infancy.

Commercialization and scalability constraints: Even when multiomics identifies a promising compound and the pathway to produce it, moving from the lab to real-world use faces hurdles. Cultured cells might produce milligram quantities, but scaling to kilograms for drug development is a big leap. Some plants might not be amenable to large bioreactors, or the cost of culture media could be prohibitive. There are also regulatory and safety considerations for using metabolites from genetically modified or tissue-cultured sources. These practical aspects mean that knowing how a plant produces something does not automatically solve how to manufacture it at scale, a gap often encountered when trying to commercialize findings.

Complexity of multi-herb formulations: Traditional medicine often uses whole extracts or mixtures of plants. Studying multiomics in a single species is challenging enough, but understanding synergistic effects in multi-herb formulations is an even bigger challenge. So far, most multiomics studies are on single species in isolation [90]. The question of how compounds from different plants might interact (both in a mixture and in a biological system) is largely unexplored.

Despite these challenges, the gaps also highlight opportunities. For instance, the lack of genomic data for many medicinal plants is being rapidly addressed by initiatives like the 1000 Medicinal Plants Genome project [91]. As more genomes become available, functional annotations will improve, making omics results more interpretable. On the data integration front, new computational methods and machine learning are being developed to handle complex omics datasets, offering hope for extracting clearer signals from the noise. Collaborative efforts and data sharing will help standardize methods.

Addressing the challenges of data volume, standardization, dynamic analysis, and translational research will bring us closer to a future where traditional medicinal plants are understood at a level of detail comparable to model crops, and where we can confidently engineer or optimize them for human health applications [88]. Efforts such as large-scale medicinal plant genome initiatives, development of standardized omics workflows, and integrated pharmacological testing platforms are key to overcoming current limitations and unlocking the full therapeutic potential of traditional plants.

## 7. Future Directions: Integrating Multiomics with Synthetic Biology and Metabolic Engineering

Looking ahead, one of the most exciting prospects is the fusion of multiomics insights with synthetic biology and metabolic engineering. Multiomics provides the blueprint of “who does what” in the complex assembly line of plant secondary metabolism; synthetic biology offers the toolkit to redesign or transfer that assembly line to new contexts for sustainable production or novel functionalities. Multiomics findings support the design of synthetic regulatory circuits that respond to cheap and sustainable inducers. Modular in vitro culture systems could be engineered to activate specific biosynthetic pathways upon addition of benign additives, temperature shifts, or light cues, offering an efficient and cost-effective route to controlled metabolite production.

Recent studies also suggest that the microbiome plays a more significant role in plant metabolite regulation than previously recognized. Leveraging multiomics, we can identify microbial partners that modulate biosynthesis or even deliver elicitors and pathway intermediates. Engineered endophytes could be programmed to co-regulate plant metabolism, creating a symbiotic factory model.

AI-driven predictive modeling, trained on multiomics datasets, offers the possibility to simulate pathway modifications before implementing them biologically. Genome-scale metabolic models (GEMs) can be refined with omics data to predict bottlenecks, while machine learning algorithms can propose optimal combinations of edits and elicitation strategies to maximize compound yield.

Multiomics investigates biosynthetic pathways for valuable compounds, then those pathways can be ported into more tractable systems (like yeast, bacteria, or fast-growing plant hosts) using synthetic biology. An interesting success story supporting this approach is the production of the antimalarial compound artemisinic acid in yeast, accomplished by identifying and assembling genes from *Artemisia annua* (with the help of omics-informed gene discovery) into a yeast metabolic network. In the coming years, we can expect many more plant pathways to be reconstructed in microbes. Multiomics will be the guide, by identifying all the necessary enzymes, their kinetics, and regulatory elements. Heterologous expressions of these pathways could provide a stable, controllable supply of compounds that are otherwise hard to obtain from the plant. Synthetic biology can also create chimeric pathways that do not exist in nature by mixing and matching enzymes from different species, potentially yielding novel analogues of natural products with improved pharmacological properties.

Another strategy is to use the knowledge from multiomics to improve the native producers. For example, if multiomics reveals a strong negative feedback regulation in a plant’s pathway (say, an end product that inhibits an early enzyme, or a repressor protein that limits expression of a pathway gene), one could use CRISPR/Cas9 to knock out or attenuate that inhibitory mechanism. Conversely, one could overexpress a transcription factor identified as a master regulator of a pathway to create a high-producing plant variety. This has already been attempted in a few cases: overexpressing the ORCA transcription factors in *Catharanthus* to increase vinblastine yield or knocking out competing pathways in Mentha to direct flux toward menthol. As more regulatory genes are identified through proteomics/transcriptomics, genome editing offers a precise way to optimize the plant’s metabolism. The synergy of omics and gene editing could lead to the next generation of medicinal plant cultivars that have significantly higher concentrations of desired compounds (much like how crops are bred for higher vitamin or oil content but now accelerated by gene editing).

We can also modify the in vitro cultures themselves. For instance, hairy root cultures could be genetically engineered to overexpress certain enzymes or to include entirely new biosynthetic routes. Synthetic biology provides standardized parts (promoters, terminators, selectable markers, etc.) to stably introduce new metabolic functions. A forward-looking idea is to engineer “smart” cell cultures that can regulate their metabolite production in response to sensor inputs, effectively programming the cells with synthetic gene circuits. This might involve inserting synthetic promoters that respond to cheap additives or environmental cues to turn on the pathway only when needed. To design such circuits, one needs detailed knowledge of promoters and signaling obtained from omics studies. Multiomics will inform which native promoters are strongly inducible and what transcriptional control points exist, which synthetic biologists can then exploit to construct controllable systems.

As the data grows, computational models of plant metabolic networks (genome-scale metabolic models) can be built and refined with omics data. These models can predict what happens if we tweak certain genes or add certain precursors. In the future, coupling these models with machine learning trained on multiomics datasets could allow predictive design: e.g., given a target compound, the algorithms might suggest which combination of gene edits and culture conditions would maximize its production [88]. Already, there is progress in using AI to mine genomics and metabolomics data for discovering biosynthetic gene clusters and regulatory motifs [92].

Multiomics does not only help in making more of existing compounds; it can highlight gaps in a pathway that could be filled to produce new-to-nature products. For example, if a particular enzyme accepts multiple substrates, one could feed or biosynthesize an alternate substrate to obtain a novel derivative. Synthetic biology could introduce that substrate’s production upstream, essentially diversifying the chemical output. This approach is akin to combinatorial chemistry but achieved biologically, sometimes called combinatorial biosynthesis. The rational basis for it often comes from knowing enzyme specificities and metabolite pools, which multiomics can provide.

An emerging angle is recognizing the role of the plant microbiome in the production of certain metabolites (some endophytes contribute genes or modulators of plant metabolism). In the future, synthetic biology might be applied not only to the plant or a microbe alone. For example, engineering a symbiotic bacterium that lives with the plant to secrete an elicitor continuously, or to take up a plant precursor and convert it to a more active compound, handing it back to the plant. This is speculative, but multiomics studies have started to show that endophytes can impact metabolite profiles of medicinal plants [93]. Harnessing this through engineered symbiosis could be a novel frontier.

All these future directions are underpinned by the advances in multiomics. However, effectively applying synthetic biology to medicinal plant pathways is identified as a challenge that needs to be addressed with innovative approaches [94]. It requires close collaboration between the data generators (omics scientists) and the bioengineers (synthetic biologists) to ensure that the right targets are chosen, and the right expression systems are used.

Encouragingly, the rapid progress in DNA synthesis and editing means we are less constrained technically, we can more freely design and test genetic constructs. The bottleneck is increasingly our understanding of the system, precisely where multiomics is most valuable. As that understanding grows, metabolic engineering can transition from a trial-and-error art to a rational design science for medicinal compounds.

The future of revisiting medicinal plants lies in not just observing nature has done but leveraging that knowledge to create what nature could do under our guidance. By uniting multiomics insights with the creative power of synthetic biology, we can imagine a future where important medicinal molecules are produced efficiently in bioreactors (either via engineered microbes or optimized plant cells), where perhaps entirely new healing compounds are bio-designed by tweaking natural pathways, and where the cultivation of medicinal plants can be more targeted and sustainable (growing plants optimized for yield of drug precursors, for example). This integrative approach promises to unlock new levels of bioactive potential far beyond what traditional practices or single-discipline science could achieve alone.

## 8. Conclusions

The integration of multiomics approaches has revolutionized the study of traditional medicinal plants, offering deeper insight into their bioactive potential. Metabolomics and proteomics have enabled precise identification of phytochemicals and elucidation of biosynthetic pathways, particularly under elicitation conditions. In vitro systems, enhanced by targeted elicitors, serve as effective platforms for controlled metabolite production. Multiomics not only bridges molecular data with biological activity but also provides a system-level understanding essential for optimizing phytochemical outputs. This interdisciplinary advancement sets the stage for future applications in synthetic biology and metabolic engineering, promoting sustainable production and novel therapeutic development. Ultimately, by aligning traditional knowledge with cutting-edge science, we pave the way for a new era of plant-based innovation.

## Figures and Tables

**Figure 1 plants-14-02029-f001:**
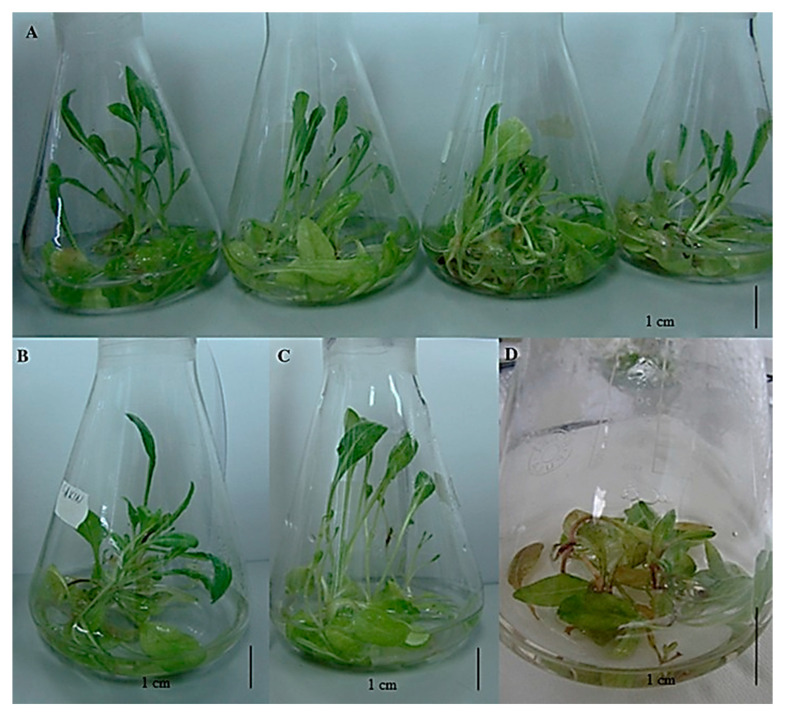
Example of shoot of Sarajevos widow flower (*Knautia sarajevensis*) cultures grown under different cytokinin concentration. (**A**) Media containing ZEA: from left to right 0, 1; 1; 2, 4 mg/L; (**B**) media containing 1 mg/L KIN; (**C**) media containing 4 mg/L KIN; (**D**) media containing no plant growth regulators [75].

**Figure 2 plants-14-02029-f002:**
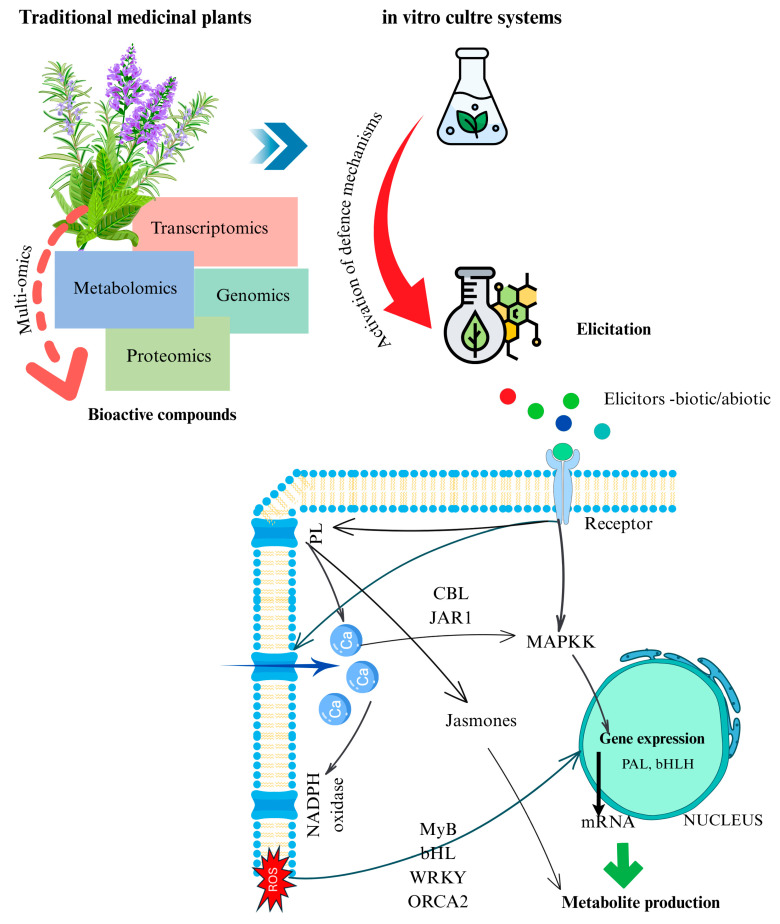
Integration of traditional medicinal plant knowledge with in vitro culture systems and elicitor-induced signaling to enhance bioactive compound production. Multiomics approaches (transcriptomics, proteomics, metabolomics, genomics) enable the identification of key biosynthetic targets. In vitro elicitation with biotic or abiotic factors activates defense pathways involving Ca^2+^ influx, MAPK cascades, ROS production, and jasmonate signaling. These responses induce transcription factors (e.g., MYB, bHLH, WRKY, ORCA2) and biosynthetic genes (e.g., *PAL*), leading to increased metabolite accumulation.

**Table 1 plants-14-02029-t001:** Examples of upregulation of protein synthesis by elicitors.

Plant Species	Elicitor	Targeted Pathway	Upregulated Proteins/Enzymes
St. John’s wort (*Hypericum perforatum*) [24,30]	*Agrobacterium* sp.	Xanthone/flavonoid	Xanthone synthase, PAL, PR proteins, glutaredoxins
Asian ginseng (*Panax ginseng*) [24,37]	Methyl jasmonate (MeJA)	Ginsenoside	Oxidosqualene cyclase (OSC), HSPs, SAMS
English jew (*Taxus baccata*) [24,38]	Methyl jasmonate (MeJA)	Taxol biosynthesis	10-deacetylbaccatin III-10-O-acetyltransferase, T5αH, PR proteins
Indian ginseng (*Withania somnifera*) [24,39]	Salicylic acid (SA)	Withanolide synthesis	HSP70, CYP450, SQS, SGT
Purple coneflower (*Echinacea purpurea*) [24,40]	Yeast extract	Caffeic acid derivatives	Caffeoyl-CoA O-methyltransferase, PAL
*Liquorice* (*Glycyrrhiza glabra*) [24,41]	Methyl jasmonate (MeJA)	Glycyrrhizin biosynthesis	β-amyrin synthase, CYP88D6, UGT73C11
Asiatic pennywort (*Centella asiatica*) [24,42]	Methyl jasmonate (MeJA)	Triterpenoid saponin	β-amyrin synthase, OSC, UGTs

**Table 2 plants-14-02029-t002:** Examples of in vitro culture systems for bioactive compound production by elicitation.

Culture Type	Plant Species	Bioactive Compound(s)	Elicitors Used
Callus cultures	Indian ginseng *Withania somnifera* [39]	Withanolides	Induced from leaf explants; MeJA and SA enhanced production
	Happy tree *Camptotheca acuminata* [41]	Camptothecin	Callus derived from leaf explants; responsive to elicitation
	English jew *Taxus baccata* [38]	Taxanes (e.g., paclitaxel)	Callus used as starting point for suspension cultures
Cell suspension cultures	Common grape wine *Vitis vinifera* [40]	Resveratrol, stilbenes	Elicited with MeJA + cyclodextrins
	Madagascar periwinkle *Catharanthus roseus* [33]	Ajmalicine, serpentine	Scalable production in bioreactors; MeJA-induced
	Maidenhair tree *Ginkgo biloba* [57]	Ginkgolides, bilobalide	Treated with salicylic acid to enhance yields
Hairy root cultures	Asian ginseng *Panax ginseng* [37]	Ginsenosides	Stable high-yield system; strong response to MeJA
	Egyptian henbane *Hyoscyamus muticus* [58]	Tropane alkaloids (e.g., scopolamine)	Induced with *A. rhizogenes*; SA and yeast extract effective
	White flax *Linum album* [59]	Podophyllotoxin	Hairy roots show higher yields than callus or suspension
Organ cultures	Sweet sagewort *Artemisia annua* (shoots) [60]	Artemisinin	Shoot cultures retain leaf-specific biosynthesis
	Sarajevos’ widow flower *Knautia sarajevensis* (shoots) [42]	Phenolics, flavonoids	SA elicitation enhanced antioxidant compounds
	Red sage *Salvia miltiorrhiza* (roots) [61]	Tanshinones	Root-derived compounds induced by bacterial co-culture

**Table 3 plants-14-02029-t003:** Elicitors, their biological effects, and representative applications in medicinal plants.

Elicitor Type	Elicitor	Mode of Action/Induced Response	Examples
Biotic	Yeast extract [42]	Induces phenylpropanoid pathway, increases phenolic/flavonoid content and antioxidant capacity	Sarajevos widow flower (*Knautia sarajevensis*): phenolics; Purple coneflower (*Echinacea purpurea*)
	*Agrobacterium rhizogenes* [72]	Triggers defense responses and secondary metabolite biosynthesis via T-DNA integration and transformation	*St. John’s wort* (*Hypericum perforatum*): xanthones, flavonoids
	Fungal cell wall fragments (e.g., chitin, glucans) [24,64]	Elicits PR proteins and oxidative stress-related secondary metabolites	General in vitro applications; Madagascar periwinkle (*Catharanthus roseus*)
	Bacterial co-culture (e.g., *Bacillus* sp.) [73]	Activates multiple stress-responsive biosynthetic pathways, enhancing compounds like tanshinones	Red sage (*Salvia miltiorrhiza*): tanshinones
Abiotic	Methyl jasmonate (MeJA) [24,37,77]	Activates jasmonate signaling cascade, boosting alkaloids, terpenoids, and phenolics	Asian ginseng (*Panax ginseng*): ginsenosides, English jew (*Taxus baccata*): paclitaxel, Asiatic pennywort (*Centella asiatica*): triterpenoids
	Salicylic acid (SA) [24,42,57]	Induces systemic acquired resistance (SAR), increases pathogenesis-related proteins and phenolic production	Maidenhair tree (*Ginkgo biloba*): bilobalide, *Knautia sarajevensis*
	Cytokinins (e.g., zeatin, BA, kinetin) [75]	Stimulates organogenesis and enhances biosynthesis of selected phenolics and phytohormones	*Knautia sarajevensis*: rosmarinic acid, flavonoids
	Nanoparticles (e.g., gold NPs, silver NPs) [26]	Enhances cytotoxicity and metabolite accumulation through ROS generation and enzyme activation	Bitter kola (*Garcinia kola*): gold NPs; general examples in callus cultures
	Heavy metals (low dose Cd, etc.) [25]	Mimics abiotic stress signals and induces oxidative defense-related pathways	*Hypericum perforatum*; general use in priming
	Physical stress (UV, temperature, osmotic) [79]	Triggers reactive oxygen species (ROS), calcium influx, MAPK signaling, and defense gene activation	Santa Catalina indian Paintbrush (*Castilleja tenuiflora*); lavender (*Lavandula angustifolia*); various in vitro systems

## Data Availability

Not applicable.

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
