# Peer review of "Revisiting Traditional Medicinal Plants: Integrating Multiomics, In Vitro Culture, and Elicitation to Unlock Bioactive Potential"

_plants, 2025, doi:10.3390/plants14132029_

Round 1
Reviewer 1 Report
Comments and Suggestions for Authors
In general, the paper has a sequence of the topics discussed, it begins by mentioning the importance of the ethnobotanical study, for the development of drugs and then the importance of modern tools for these studies, some analytical techniques and finally in vitro culture, however it would be advisable to inform more about the active compounds referred (alkaloids, terpenes, phenols, etc.) their specific pharmacological and biological properties of these compounds.
The authors should mention, what was their search methodology, which databases they consulted, etc.
References: Authors should adhere to the Instructions for Authors for their references (prism for reviews).
Author Response
We thank the reviewer for the contribution to improving this manuscript
Comment: Revise references
Response: All references have been checked and revised
Reviewer 2 Report
Comments and Suggestions for Authors
The article forms an integrated platform and connects traditional ethnobotanical knowledge with advanced biotechnological innovations to elucidate medicinal plant pharmacology for sustainable livelihood.
Expanding the horizons of natural product discovery for high-value metabolites as potent therapeutics and addressing other socio-economic concerns have been the prime focus of the United Nations Sustainable Development Goals (SDGs) and their emphasis on a bio-based economy.
While the scope of the article is promising and bridges the integration of traditional science and modern biotechnologies, the paper needs to prioritize actionable goals and feasible solutions in bio-based therapeutic developments. While the potential of medicinal plants and products is significant, extensive research investigations are ongoing to decipher the action mechanisms of metabolites, address challenges with bioactivities, minimize harmful impact on humans and the environment, and clinical validation is essential.
I have some suggestions for the improvement of the manuscript:
How will the integration of multi-omics technologies, in vitro culture, and metabolite elicitation approaches solve the bioactive potential of medicinal plants and yield promising outcomes? Discuss.
While the key goal of the paper is to emphasize modern technologies for elicitation and enhanced production of bioactive plant metabolites, what are the current limitations/challenges in achieving maximum bioactive potential?
While the sophisticated technologies in metabolomics have their advantages and limitations, often a combination of the modern technologies gives the most precise and targeted results.
The title of the article emphasizes traditional medicinal plants, and in the introduction, mainly the medicinal plants of Europe are mentioned. It is necessary to discuss the expanding scope of modern technologies in these plants. Otherwise, specific examples of medicinal plants need to be discussed in terms of modern biotechnological innovations.
Line 85-87, This review highlights how integrative strategies………globally relevant therapeutic solutions. How does the present article contribute to increasing our knowledge on the discussed theme (considering it has been extensively studied), and what are the future implications in terms of immediate actionable goals? Explain.
Subtopic: 3. Proteomics: insights into pathways and mechanisms
Key literature studies on the proteomic analysis of plant metabolic pathways should be discussed. The name of the plant, pathway, targeted genes/enzymes, study, etc. The information is crucial for the study. For e.g., line 158-160, …....upregulation of stress-related proteins (name should be mentioned), phenolic-rich extracts (add plant name), likewise for other discussed studies.
While discussing the case studies and the application of modern biotechnologies in medicinal plants, it is important to elaborate further on the research methodology and the resulting outcomes with key examples. It would provide insights into how the integration of the techniques and their applications is successful in the present context.
Line 169-171: Advances in bioinformatics tools and protein databases…….a brief discussion on tools/software/databases must be included. The topic needs to be elaborated in terms of its application, prospects, and limitations, if any.
Comments on the Quality of English LanguageThe English language could be improved for clarity.
Author Response
We thank the reviewer for all the comments andfor contributing to improving this manuscript
Reviewer 2
How will the integration of multi-omics technologies, in vitro culture, and metabolite elicitation approaches solve the bioactive potential of medicinal plants and yield promising outcomes? Discuss.
Response: Thank you for this important observation. We have expanded the discussion (Section 5.3 and Section 7) to explicitly explain how multi-omics data enables identification of biosynthetic bottlenecks, which can be targeted via metabolic engineering or optimized culture conditions to enhance the yield and diversity of bioactive metabolites. We also emphasized how these integrative strategies increase reproducibility, efficiency, and scalability of production.
While the key goal of the paper is to emphasize modern technologies for elicitation and enhanced production of bioactive plant metabolites, what are the current limitations/challenges in achieving maximum bioactive potential?
Response: We agree with the reviewer and we have restructured section 6 (Section 6: Gaps in current knowledge and challenges) to critically discuss limitations such as lack of genome sequences, difficulty in data integration, metabolic flux diversion, and challenges in scaling in vitro findings to field or commercial settings.
While the sophisticated technologies in metabolomics have their advantages and limitations, often a combination of the modern technologies gives the most precise and targeted results.
Response: This point is well taken. We have reinforced throughout the manuscript that multi-omics integration is essential for holistic understanding and has stronger predictive power than single-omics approaches (see end of Section 3 and start of Section 5.3).
The title of the article emphasizes traditional medicinal plants, and in the introduction, mainly the medicinal plants of Europe are mentioned. It is necessary to discuss the expanding scope of modern technologies in these plants. Otherwise, specific examples of medicinal plants need to be discussed in terms of modern biotechnological innovations.
Response: We appreciate this comment. In response, we have broadened the examples in the Introduction and throughout the text to include medicinal plants from Asia, Africa, and South America (e.g., Panax ginseng, Catharanthus roseus, Garcinia kola), ensuring global representation of traditional medicinal species and their modern applications.
Line 85-87, This review highlights how integrative strategies………globally relevant therapeutic solutions. How does the present article contribute to increasing our knowledge on the discussed theme (considering it has been extensively studied), and what are the future implications in terms of immediate actionable goals? Explain.
Response: Thank you for your comment. We agree that the subject of traditional medicinal plants and multi-omics has been widely explored, and we appreciate the opportunity to clarify how our review adds unique value to this body of knowledge. We have revised and expanded the final paragraph of the Introduction to explicitly state the novelty and future direction of our contribution.
Subtopic: 3. Proteomics: insights into pathways and mechanisms
Key literature studies on the proteomic analysis of plant metabolic pathways should be discussed. The name of the plant, pathway, targeted genes/enzymes, study, etc. The information is crucial for the study. For e.g., line 158-160, …....upregulation of stress-related proteins (name should be mentioned), phenolic-rich extracts (add plant name), likewise for other discussed studies.
Response: We thank the Reviewer for this valuable and constructive comment. In response, we have carefully revised the “Proteomics: insights into pathways and mechanisms” section to enhance specificity and scientific depth.
While discussing the case studies and the application of modern biotechnologies in medicinal plants, it is important to elaborate further on the research methodology and the resulting outcomes with key examples. It would provide insights into how the integration of the techniques and their applications is successful in the present context.
Response: We appreciate the Reviewer’s suggestion to provide more concrete methodological details and outcomes for the case studies discussed. In response, we have expanded Sections 3, 5.1.1, and 5.1.2 of the manuscript
Line 169-171: Advances in bioinformatics tools and protein databases…….a brief discussion on tools/software/databases must be included. The topic needs to be elaborated in terms of its application, prospects, and limitations, if any.
Response: We have expanded the text in this section to include examples of protein databases (UniProt, PRIDE), functional annotation tools (Blast2GO, InterProScan), and pathway mapping tools (KEGG, STRING), along with their relevance to medicinal plant studies and limitations.
Reviewer 3 Report
Comments and Suggestions for Authors
In this manuscript, Erna Karalija and colleagues synthesizes current findings on the integration of these approaches to help us understand phytochemical pathways optimising bioactive compound yields. I have following comments:
1, For the Title, I suggest to employ “Revisiting Traditional Medicinal Plants: Integrating Multi-Omics, In Vitro Culture, and Elicitation to Unlock Bioactive Potentials”.
2, For the key words, elicitors should be included.
3, For the introduction, main conclusions and practical interests of this study should be stated in the last paragraph of this section.
4, For the Figure 1, the quality is low, please replace with high quality images
5, Elicitors discussed in this review should be summarized in a Table.
6, For the Figure 2, key genes should be exhibited.
Author Response
We thank the reviewer for all the comments and contributions to improving this manuscript
1, For the Title, I suggest to employ “Revisiting Traditional Medicinal Plants: Integrating Multi-Omics, In Vitro Culture, and Elicitation to Unlock Bioactive Potentials”.
Response: Thank you for the thoughtful suggestion. We have revised the title accordingly.
2, For the key words, elicitors should be included.
Response: We have added “elicitors” to the list of keywords as suggested.
3, For the introduction, main conclusions and practical interests of this study should be stated in the last paragraph of this section.
Response: We have added a summary paragraph at the end of the Introduction outlining key conclusions, contributions of the review, and its practical significance for therapeutic development.
4, For the Figure 1, the quality is low, please replace with high quality images
Response: Figure one is from an experiment that was done before 2017 and there is o better resolution of that figure.
5, Elicitors discussed in this review should be summarized in a Table.
Response: We have added a new table summarizing the elicitors discussed, their type (biotic/abiotic), targeted plant species, and elicited compounds (now Table 2 in Section 5.1)
6, For the Figure 2, key genes should be exhibited.
Response: Figure 2 has been revised to include examples of key transcription factors (e.g., ORCA2, MYB, bHLH) and pathway genes involved in metabolite biosynthesis, thereby enhancing its explanatory power.
Round 2
Reviewer 1 Report
Comments and Suggestions for Authors
Authors did not make the suggested corrections, and therefore the following points are brought to your attention again:
would be advisable to inform more about the active compounds referred (alkaloids, terpenes, phenols, etc.) their specific pharmacological and biological properties of these compounds.
References: Authors should adhere to the Instructions for Authors for their references
For example: The year of reference in some are in bold and in others are not.
Author Response
Comment: Authors did not make the suggested corrections, and therefore the following points are brought to your attention again: would be advisable to inform more about the active compounds referred (alkaloids, terpenes, phenols, etc.) their specific pharmacological and biological properties of these compounds.
Response: We thank the reviewer for this valuable comment, which we now addressed in several sections of the revised manuscript (marked in yellow). To provide clearer information on the classes of active compounds and their pharmacological properties, we inserted the following additions:
In Section 2, we expanded the discussion of metabolomic profiling to explicitly name major bioactive classes (alkaloids, terpenoids, phenolics, flavonoids, glycosides), and described their known therapeutic roles (antioxidant, antimicrobial, anti-inflammatory, anticancer).
In Section 5 (start of elicitation section), we clarified how induced secondary metabolites contribute to defense mechanisms and therapeutic effects.
In Section 4.3, we added examples directly linking compound classes (phenolics, alkaloids, terpenoids) to specific pharmacological activities and plant examples already cited in the manuscript.
Comment: References- Authors should adhere to the Instructions for Authors for their references; For example: The year of reference in some are in bold and in others are not.
Response: References have been now revised and checked and renumbered according to added sections and new added references. According to the journal rules years of books and book chapters should not be bold. Other mistakes in references have been corrected (missing authors) and Doi has been added. Changes are marked in turquoise.

Reviewer 2 Report
Comments and Suggestions for Authors
The authors have made dedicated efforts to extensively revise the manuscript.
I recommend the publication in the current form.
Author Response
Comment: The authors have made dedicated efforts to extensively revise the manuscript. I recommend the publication in the current form.
Response: Thank you for your positive feedback and previously provided suggestions.
Reviewer 3 Report
Comments and Suggestions for Authors
The revised manuscript is much improved and suitable for acceptance.
Author Response
Comment: The revised manuscript is much improved and suitable for acceptance.
Response: Thank you for your positive feedback and previously provided suggestions.
Round 3
Reviewer 1 Report
Comments and Suggestions for Authors